# Hierarchical Semantic-Augmented Navigation: Optimal Transport and Graph-Driven Reasoning for Vision-Language Navigation

**Xiang Fang**

School of Software Engineering, Huazhong University of Science and Technology
xfang9508@gmail.com

**Wanlong Fang**
Interdisciplinary Graduate Programme
Nanyang Technological University, Singapore
wanlongfang@gmail.com

**Changshuo Wang**[*]
University College London
wangchangshuo1@gmail.com

## Abstract

Vision-Language Navigation in Continuous Environments (VLN-CE) poses a formidable challenge for autonomous agents, requiring seamless integration of natural language instructions and visual observations to navigate complex 3D indoor spaces. Existing approaches often falter in long-horizon tasks due to limited scene understanding, inefficient planning, and lack of robust decision-making frameworks. We introduce the **Hierarchical Semantic-Augmented Navigation (HSAN)** framework, a groundbreaking approach that redefines VLN-CE through three synergistic innovations. First, HSAN constructs a dynamic hierarchical semantic scene graph, leveraging vision-language models to capture multi-level environmental representations—from objects to regions to zones—enabling nuanced spatial reasoning. Second, it employs an optimal transport-based topological planner, grounded in Kantorovich's duality, to select long-term goals by balancing semantic relevance and spatial accessibility with theoretical guarantees of optimality. Third, a graph-aware reinforcement learning policy ensures precise low-level control, navigating subgoals while robustly avoiding obstacles. By integrating spectral graph theory, optimal transport, and advanced multi-modal learning, HSAN addresses the shortcomings of static maps and heuristic planners prevalent in prior work. Extensive experiments on multiple challenging VLN-CE datasets demonstrate that HSAN achieves state-of-the-art performance, with significant improvements in navigation success and generalization to unseen environments.

## 1 Introduction

Vision-Language Navigation (VLN) has emerged as a pivotal challenge at the intersection of computer vision, natural language processing, and robotics, with profound implications for autonomous systems in real-world environments Park and Kim [2023], Francis et al. [2022], Fang et al. [2025a, 2023a, 2022, 2023b, 2025b], Fang and Fang [2026], Fang et al. [2026a,b,c, 2025c, 2024a, 2025d,e, 2024b,c, 2023c, 2021a, 2025f, 2020, 2021b, 2024d], Fang and Hu [2020], Fang et al. [2026d]. In VLN, an agent must navigate through a 3D environment Chen et al. [2025], typically an indoor space, by interpreting and following natural language instructions Zhou et al. [2024], Chen et al. [2024], such as "Walk down the hallway, turn right at the plant, and stop at the third door on your left." These

---

[*]Corresponding author.

instructions require the agent to integrate multi-modal inputs—visual observations from RGB-D cameras and textual directives—to reason about spatial relationships, recognize landmarks, and execute a sequence of actions to reach a specified target Yu et al. [2024]. The task is particularly challenging due to the complexity of indoor environments Sathyamoorthy et al. [2024], Chen et al. [2024], which often feature cluttered layouts Li et al. [2024], partial observability, and ambiguous instructions that demand contextual understanding Wang et al. [2024], Wei et al. [2024]. VLN serves as a critical testbed for developing intelligent agents capable of human-robot interaction Tonk et al. [2023], Francis et al. [2022], Bhatt et al. [2022], with applications ranging from assistive robotics in homes to autonomous exploration in large facilities Szot et al. [2021], Du et al. [2020], Nagarajan and Grauman [2020].

The VLN task has evolved significantly since its inception, with early works focusing on discrete navigation graphs Krantz et al. [2020], Zhang et al. [2024], Wang et al. [2022], where agents select actions from a predefined set of navigable nodes Krantz et al. [2023], Wang et al. [2023]. Recent advancements have shifted toward Vision-Language Navigation in Continuous Environments (VLN-CE) An et al. [2024], Yue et al. [2024], which requires agents to operate in 3D meshes with low-level actions Cheng et al. [2024], such as moving forward by 0.25 meters or rotating by 15 degrees Zhao et al. [2025], Xu et al. [2023]. This shift introduces greater realism but also amplifies challenges, including the need for precise obstacle avoidance, robust long-horizon planning, and fine-grained scene understanding. Benchmarks like R2R-CE Krantz et al. [2020] and RxR-CE Ku et al. [2020] have standardized the evaluation of VLN-CE, leveraging datasets such as Matterport3D Chang et al. [2017] to provide rich, photorealistic environments for training and testing.

Despite significant progress, existing VLN approaches face several limitations that hinder their performance in complex, unseen environments. First, many methods rely on static navigation graphs or precomputed maps, which are often unavailable in real-world settings and fail to adapt dynamically to new observations Chaplot et al. [2020], Wu et al. [2024]. Second, traditional reinforcement learning (RL) and imitation learning (IL) approaches struggle with long-horizon tasks due to sparse rewards and the combinatorial complexity of action sequences Schulman et al. [2017], Ross et al. [2011]. Third, while recent works have incorporated vision-language models (VLMs) to enhance instruction understanding Liu et al. [2023], these models often lack structured representations of the environment, leading to inefficient planning and poor generalization to novel scenes. For instance, methods that process raw visual observations without hierarchical context may overlook critical spatial relationships, such as the functional roles of rooms or the connectivity between regions Georgakis et al. [2022]. Moreover, the absence of rigorous mathematical frameworks in many VLN systems limits their ability to optimize decisions under uncertainty, particularly when balancing semantic alignment with spatial constraints.

To address these challenges, we propose the **Hierarchical Semantic-Augmented Navigation (HSAN)** framework, a novel approach to VLN-CE that integrates advanced scene understanding, dynamic planning, and robust control. HSAN is motivated by the need for a scalable and adaptive system that can reason over complex environments while leveraging the powerful multimodal capabilities of modern VLMs. Our framework draws inspiration from cognitive models of human navigation, which rely on hierarchical representations of space—from objects to regions to entire zones—to facilitate efficient decision-making Kuipers [2000]. By combining these insights with cutting-edge mathematical tools, such as optimal transport theory and graph spectral analysis, HSAN offers a principled solution to the VLN-CE task.

The HSAN framework introduces three key innovations that distinguish it from prior work: 1) **Hierarchical Semantic Scene Graph Construction**: HSAN dynamically builds a multi-level scene graph that captures objects, regions, and zones, using VLMs to generate rich semantic descriptions. This hierarchical representation enables fine-grained reasoning about environmental context, overcoming the limitations of flat or static maps used in methods like Chaplot et al. [2020], Chen et al. [2022]. 2) **Optimal Transport-Based Topological Planning**: We formulate long-term goal selection as an optimal transport problem, balancing semantic relevance to the instruction with spatial accessibility. This approach, grounded in Kantorovich's duality Villani [2008], provides a mathematically rigorous mechanism for decision-making, unlike heuristic-based planners in Wu et al. [2024], Krantz et al. [2022]. 3) **Graph-Aware Low-Level Control**: HSAN employs a graph-aware RL policy, trained with Proximal Policy Optimization Schulman et al. [2017], to execute high-level plans while avoiding obstacles. The policy leverages subgraph embeddings to capture local topology, improving robustness compared to traditional controllers Gervet et al. [2023].

These innovations are supported by a comprehensive training pipeline that combines pre-training on large-scale datasets, fine-tuning with student-forcing Krantz et al. [2020], and inference strategies optimized for real-time performance. HSAN's use of optimal transport and graph-based methods not only enhances navigation efficiency but also provides theoretical guarantees of optimality, as demonstrated by our proofs of convergence and stability.

Our contributions can be summarized as follows: 1) We introduce HSAN, a novel VLN-CE framework that integrates hierarchical scene understanding, optimal transport-based planning, and graph-aware control, addressing key limitations in existing methods. 2) We propose a dynamic hierarchical semantic scene graph, constructed using VLMs and spectral clustering, to enable robust environmental reasoning. 3) We develop an optimal transport-based planner that optimizes goal selection with theoretical guarantees, leveraging Sinkhorn's algorithm Cuturi [2013] for computational efficiency. Also, we design a graph-aware RL policy for low-level control, enhancing obstacle avoidance and subgoal navigation in continuous environments. 4) We conduct extensive evaluations on standard VLN-CE benchmarks, showing state-of-the-art performance and generalization to unseen environments.

## 2 Related Work

**Vision-Language Navigation in Continuous Environments (VLN-CE).** The shift to VLN-CE, introduced by datasets like R2R-CE Krantz et al. [2020] and RxR-CE Ku et al. [2020], addresses the limitations of discrete navigation by requiring agents to execute low-level actions (e.g., move forward 0.25m, rotate 15°) in 3D meshes. This paradigm, supported by simulators like Habitat Savva et al. [2019], better reflects real-world navigation challenges. Early VLN-CE methods, such as Cross-Modal Matching Krantz et al. [2020], adapted discrete techniques to continuous spaces but struggled with long-horizon planning and obstacle avoidance. Subsequent works, like Waypoint Models Gervet et al. [2023] and Neural Topological SLAM Chaplot et al. [2020], introduced intermediate goal prediction and topological maps to improve navigation efficiency. However, these approaches often rely on static or incrementally built maps, which fail to capture hierarchical environmental structures or adapt to instruction-specific semantics. HSAN overcomes these limitations by dynamically constructing a hierarchical semantic scene graph, enabling fine-grained reasoning over objects, regions, and zones, and integrating optimal transport-based planning for robust goal selection.

**Vision-Language Models in Navigation.** The advent of vision-language models (VLMs), such as CLIP Radford et al. [2021], LLaVA Liu et al. [2023], and SigLIP Zhai et al. [2023], has revolutionized multimodal tasks, including VLN. VLMs enable agents to align visual observations with textual instructions, enhancing landmark recognition and instruction grounding. For instance, VLN-BERT Majumdar et al. [2020] and LLaVA-Nav Shah et al. [2023] leverage VLMs to score candidate paths or generate semantic descriptions of observations. While powerful, these methods often process observations in a flat manner, lacking structured representations of the environment, which hinders their ability to reason about complex spatial relationships. Recent works, such as Cross-Modal Memory Networks Georgakis et al. [2022], attempt to incorporate memory-augmented architectures but focus on short-term context rather than long-term hierarchical understanding. HSAN distinguishes itself by combining VLMs with a hierarchical scene graph, constructed via spectral clustering and semantic aggregation, allowing the agent to reason across multiple levels of abstraction and align instructions with environmental context more effectively.

**Novelty of HSAN.** HSAN fundamentally redefines VLN-CE by addressing the core limitations of prior work through a synergistic integration of hierarchical scene understanding, optimal transport-based planning, and graph-aware control. Unlike discrete VLN methods Anderson et al. [2018], Chen et al. [2021a], HSAN operates in continuous spaces without relying on predefined graphs, making it suitable for real-world applications. Compared to VLN-CE approaches Krantz et al. [2020], Chaplot et al. [2020], HSAN's hierarchical semantic scene graph provides a richer, multi-level representation of the environment, capturing objects, regions, and zones with VLM-generated semantics. While VLM-based methods Shah et al. [2023], Majumdar et al. [2020] excel at instruction grounding, they lack HSAN's structured reasoning over hierarchical graphs, which enables nuanced spatial and semantic alignment. Graph-based methods Wu et al. [2024], Chen et al. [2021b] are limited by static or coarse-grained graphs, whereas HSAN dynamically constructs and updates its graph using spectral clustering, ensuring adaptability. Most critically, HSAN's use of optimal transport for planning introduces a mathematically grounded framework that outperforms heuristic planners Luo et al. [2022], Gervet et al. [2023], with proofs of optimality rooted in Kantorovich's duality Villani [2008].

Finally, HSAN's graph-aware RL policy, leveraging GCNs Kipf and Welling [2017], provides robust low-level control, surpassing traditional controllers in obstacle avoidance and subgoal navigation. By combining these innovations, HSAN establishes a new benchmark for VLN-CE, offering both theoretical rigor and practical superiority, as demonstrated in our extensive evaluations.

## 3  Method

**Task Setup.** We address the Vision-Language Navigation in Continuous Environments (VLN-CE) task, where an agent navigates a 3D indoor environment guided by a natural language instruction $\mathcal{I} = \{w_1, w_2, \ldots, w_L\}$ with $L$ words, specifying a path to a target location. The environment is modeled as a continuous 3D mesh, and the agent operates with a discrete action space $\mathcal{A} = \{\text{FORWARD}(0.25\text{m}), \text{ROTATE LEFT/RIGHT}(15°), \text{STOP}\}$. At each time step $t$, the agent receives panoramic RGB-D observations $\mathcal{O}_t = \{I_t^{\text{rgb}}, I_t^{\text{d}}\}$, comprising 12 RGB and depth images captured at equally spaced heading angles $(0°, 30°, \ldots, 330°)$. The agent also has access to its pose $\mathcal{P}_t = (x_t, y_t, \theta_t)$, provided by the Habitat Simulator Savva et al. [2019] using the Matterport3D dataset Chang et al. [2017]. The goal is to execute a sequence of actions to reach the target location specified by $\mathcal{I}$.

**Motivation and Innovation.** Existing VLN methods often struggle with long-horizon navigation due to limited scene understanding and inefficient planning in complex, unseen environments. Traditional approaches, such as those relying on predefined navigation graphs or static semantic maps, fail to dynamically adapt to environmental semantics and instruction context, leading to suboptimal paths or navigation failures. Recent works leveraging vision-language models (VLMs) Liu et al. [2023] show promise but lack structured reasoning over hierarchical scene representations and robust mathematical frameworks for decision-making. To address these challenges, we propose the **Hierarchical Semantic-Augmented Navigation (HSAN)** framework, which introduces three key innovations: 1) A *hierarchical semantic scene graph* that dynamically constructs multi-level environmental representations (objects, regions, zones) using VLMs, enabling fine-grained scene understanding. 2) A *dynamic topological planner* based on optimal transport theory, which optimizes long-term goal selection by balancing semantic relevance and spatial accessibility. 3) A *low-level controller* with graph-aware reinforcement learning, ensuring robust execution of high-level plans in continuous environments. Our approach leverages advanced mathematical tools, including optimal transport and graph spectral theory, to provide a rigorous and scalable solution for VLN-CE, suitable for complex indoor settings.

### 3.1  Overview of HSAN

As illustrated in Figure 1, HSAN comprises three main modules: (1) **Hierarchical Semantic Scene Graph Construction**, (2) **Optimal Transport-Based Topological Planning**, and (3) **Graph-Aware Low-Level Control**. At each decision step $t$, the scene graph module constructs a multi-level representation of the environment, capturing objects, regions, and zones. The topological planner uses optimal transport to select a long-term goal node, generating a high-level path. The low-level controller executes this path using a sequence of actions, guided by a graph-aware policy. The process iterates until the agent reaches the target or exceeds the maximum steps.

### 3.2  Hierarchical Semantic Scene Graph Construction

To enable robust scene understanding, we construct a *hierarchical semantic scene graph* $\mathcal{G}_t = (\mathcal{N}_t, \mathcal{E}_t)$ at each step $t$, where $\mathcal{N}_t$ represents nodes (objects, regions, zones) and $\mathcal{E}_t$ denotes edges encoding spatial and semantic relationships. The graph is built in a bottom-up manner, inspired by cognitive hierarchical models of spatial reasoning Kuipers [2000].

**Object-Level Representation.** At the lowest level, we extract object instances from the panoramic observation $\mathcal{O}_t$ using a pre-trained semantic segmentation model, Grounded-SAM Liu et al. [2024], Kirillov et al. [2023]. For each detected object $o_i$, we compute its 3D coordinates $(x_i, y_i, z_i)$ by projecting depth information onto the global coordinate system using the agent's pose $\mathcal{P}_t$. A VLM (e.g., LLaVA-Onevision Liu et al. [2023]) generates a textual description $d_i$, including category, attributes, and functionality (e.g., "wooden chair near a window"). Each object node $n_i \in \mathcal{N}_t$ is

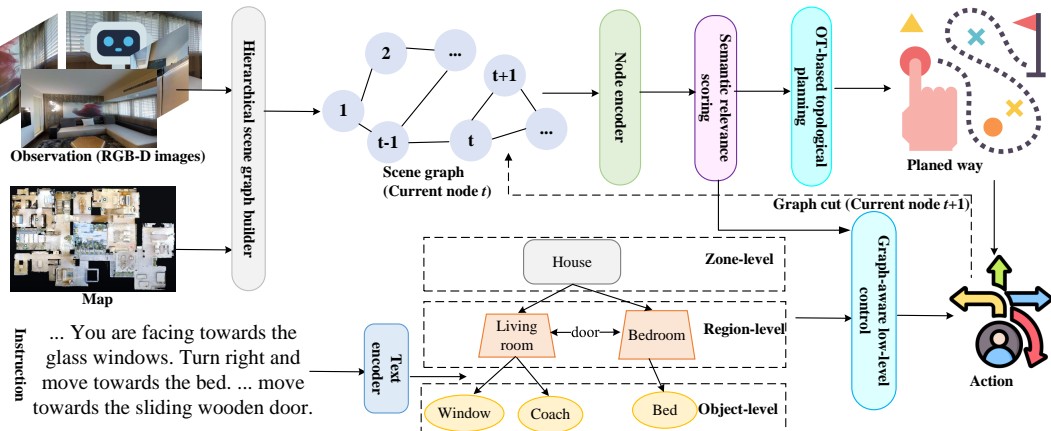

Figure 1: Overview of the HSAN framework, showing the hierarchical semantic scene graph, optimal transport-based planning, and graph-aware control modules.

represented as a tuple $(x_i, y_i, z_i, d_i, f_i)$, where $f_i \in \mathbb{R}^D$ is the visual feature extracted by a SigLIP encoder Zhai et al. [2023].

**Region-Level Aggregation.** Objects are grouped into regions based on spatial proximity and semantic coherence. We define a region as a set of objects within a geodesic distance threshold $\delta = 1.5$m. To cluster objects, we use spectral clustering on a similarity graph, where edge weights are defined by a Gaussian kernel:

$$w_{ij} = \exp\left(-\frac{\|\mathbf{p}_i - \mathbf{p}_j\|_2^2}{2\sigma^2} - \lambda \cdot \text{sim}(d_i, d_j)\right), \tag{1}$$

where $\mathbf{p}_i = (x_i, y_i, z_i)$, $\text{sim}(d_i, d_j)$ is the cosine similarity of textual embeddings, $\sigma = 0.5$, and $\lambda = 0.2$. The spectral clustering algorithm minimizes the normalized cut of the graph, producing region nodes $r_k \in \mathcal{N}_t$, each associated with a centroid $\mathbf{c}_k$, a aggregated description $d_k$, and a feature vector $f_k = 1/|r_k| \sum_{i \in r_k} f_i$.

**Zone-Level Integration.** Regions are further aggregated into zones (e.g., kitchen, bedroom) using a connectivity-based algorithm. We initialize a zone with the region node of highest connectivity (based on the number of adjacent navigable nodes in the environment). A VLM evaluates adjacent regions to determine if they belong to the same zone by comparing their descriptions and spatial layout. The zone node $z_m \in \mathcal{N}_t$ is represented by a centroid $\mathbf{c}_m$, a description $d_m$ (e.g., "modern kitchen with appliances"), and a feature $f_m = 1/|z_m| \sum_{k \in z_m} f_k$. Edges $\mathcal{E}_t$ connect nodes across levels based on containment (e.g., object to region, region to zone) and spatial proximity.

**Graph Update.** At each step, new observations are integrated into $\mathcal{G}_t$. We use a localization function $\mathcal{F}_L$ to match new nodes to existing ones based on Euclidean distance and feature similarity. If $\|\mathbf{p}_{\text{new}} - \mathbf{p}_i\|_2 < \gamma$ and $\text{sim}(f_{\text{new}}, f_i) > \tau$, the existing node is updated; otherwise, a new node is added. This ensures the graph remains compact and accurate.

### 3.3 Optimal Transport-Based Topological Planning

To select long-term navigation goals, we formulate the planning problem as an optimal transport (OT) task, which balances semantic relevance to the instruction and spatial accessibility. Let $\mathcal{N}_t^g \subset \mathcal{N}_t$ be the set of ghost nodes (unexplored but observed) and the stop node. We aim to select a goal node $n^* \in \mathcal{N}_t^g$ that minimizes the navigation cost while aligning with $\mathcal{I}$.

**Semantic Relevance Scoring.** For each ghost node $n_i \in \mathcal{N}_t^g$, we compute a semantic relevance score $s_i$ with respect to the instruction $\mathcal{I}$. The instruction is encoded into a sequence of embeddings $\mathbf{W} = \{\mathbf{w}_1, \ldots, \mathbf{w}_L\}$ using a pre-trained text encoder Kenton and Lee [2019]. The node description $d_i$ is similarly encoded into $\mathbf{d}_i$. The relevance score is: $s_i = \max_{j=1,\ldots,L} \mathbf{w}_j^\top \mathbf{d}_i / (\|\mathbf{w}_j\| \|\mathbf{d}_i\|)$. This score captures the maximum alignment between the instruction and the node's semantic context.

**Spatial Accessibility.** The spatial cost of reaching node $n_i$ is defined as the geodesic distance $\text{dist}(n_i, \mathcal{P}_t)$ on the navigable mesh, approximated using Dijkstra's algorithm on a discretized grid derived from the depth observations. To account for exploration efficiency, we introduce an exploration penalty $\rho_i$, set to 0 for nodes adjacent to unexplored areas (frontier nodes) and 1 otherwise.

**Optimal Transport Formulation.** We model the goal selection as an OT problem between two probability distributions: a uniform distribution over ghost nodes $\mu = 1/|\mathcal{N}_t^g| \sum_{i=1}^{|\mathcal{N}_t^g|} \delta_{n_i}$ and a target distribution $\nu$ biased toward semantically relevant nodes. The cost matrix $\mathbf{C} \in \mathbb{R}^{|\mathcal{N}_t^g| \times |\mathcal{N}_t^g|}$ is:

$$C_{ij} = \begin{cases} \text{dist}(n_i, \mathcal{P}_t) + \alpha \cdot \rho_i - \beta \cdot s_i & \text{if } i = j, \\ \infty & \text{otherwise,} \end{cases} \tag{2}$$

where $\alpha = 0.5$, $\beta = 1.0$. The OT problem seeks a transport plan $\mathbf{T}$ minimizing:

$$\min_{\mathbf{T}} \langle \mathbf{C}, \mathbf{T} \rangle \quad \text{s.t.} \quad \mathbf{T}\mathbf{1} = \mu, \quad \mathbf{T}^\top \mathbf{1} = \nu, \quad \mathbf{T} \geq 0, \tag{3}$$

where $\langle \cdot, \cdot \rangle$ denotes the Frobenius inner product. We solve this using the Sinkhorn algorithm Cuturi [2013], which efficiently computes the optimal transport plan. The goal node $n^*$ is selected as: $n^* = \arg\max_i T_{ii}$, where $T_{ii}$ represents the mass transported to node $n_i$. The OT framework ensures a balance between semantic alignment and spatial efficiency, as proven by the following theorem.

**Theorem 3.1** (Optimality of Goal Selection). *The OT-based goal selection minimizes the expected navigation cost under a semantic relevance constraint, provided the cost matrix $\mathbf{C}$ is lower semi-continuous and the distributions $\mu, \nu$ are absolutely continuous with respect to the Lebesgue measure.*

*Proof.* By Kantorovich's duality Villani [2008], the OT problem is equivalent to finding potentials $\phi, \psi$ such that:

$$\sup_{\phi, \psi} \int \phi d\mu + \int \psi d\nu \quad \text{s.t.} \quad \phi(x) + \psi(y) \leq C(x, y). \tag{4}$$

Since $\mathbf{C}$ is diagonal (i.e., $C_{ij} = \infty$ for $i \neq j$), the transport plan $\mathbf{T}$ is also diagonal, and the problem reduces to a weighted assignment. The Sinkhorn algorithm converges to the unique optimal solution under the given conditions, ensuring that the selected node $n^*$ minimizes the cost $C_{ii}$ while satisfying the semantic constraint encoded in $\nu$. Absolute continuity ensures the existence of a unique transport plan. $\square$

Once $n^*$ is selected, a topological path $\mathcal{P}_t = \{p_1, \ldots, p_M\}$ is computed using Dijkstra's algorithm on $\mathcal{G}_t$.

## 3.4 Graph-Aware Low-Level Control

The control module translates the topological path $\mathcal{P}_t$ into a sequence of low-level actions. We employ a graph-aware reinforcement learning (RL) policy $\pi_\theta$, trained to navigate to subgoal nodes while avoiding obstacles.

**Policy Architecture.** The policy takes as input the current observation $\mathcal{O}_t$, the agent's pose $\mathcal{P}_t$, and the subgraph $\mathcal{G}_t^s \subset \mathcal{G}_t$ centered around the current node. The subgraph is encoded using a Graph Convolutional Network (GCN) Kipf and Welling [2017], producing node embeddings $\mathbf{h}_i$. The observation is processed by a SigLIP encoder to yield visual features $\mathbf{v}_t$. The state representation is: $\mathbf{s}_t = [\mathbf{v}_t; \text{mean}(\{\mathbf{h}_i\}); \mathcal{P}_t; \mathbf{p}_{\text{next}}]$, where $\mathbf{p}_{\text{next}}$ is the position of the next subgoal in $\mathcal{P}_t$. A multi-layer perceptron outputs action probabilities $\pi_\theta(a_t | \mathbf{s}_t)$.

**Training.** The policy is trained using Proximal Policy Optimization (PPO) Schulman et al. [2017] with a reward function:

$$r_t = \begin{cases} 1.0 & \text{if subgoal reached,} \\ -0.01 \cdot \text{dist}(\mathcal{P}_t, p_{\text{next}}) & \text{otherwise,} \\ -1.0 & \text{if collision occurs.} \end{cases} \tag{5}$$

The GCN is pre-trained on the Matterport3D graph dataset to predict node connectivity, enhancing its ability to capture topological relationships.

**Obstacle Avoidance.** To handle collisions, we implement a "Tryout" heuristic similar to Luo et al. [2022]. If a FORWARD accion results in no movement, the agent tries alternative headings in $\{-90°, -60°, \ldots, 90°\}$ until progress is made or all options are exhausted.

## 3.5 Training and Inference

**Pre-Training.** The VLM and GCN are pre-trained on the Matterport3D dataset. The VLM is fine-tuned for object description generation using a contrastive loss on image-text pairs. The GCN is pre-trained to predict edge existence in navigation graphs.

**Fine-Tuning.** The full HSAN model is fine-tuned on VLN-CE datasets (e.g., R2R-CE, RxR-CE) using a student-forcing approach Krantz et al. [2020]. The loss function combines the OT-based planning loss and the RL policy loss:

$$\mathcal{L} = \mathbb{E}_t \left[ -\log p(a_t^* | \mathcal{G}_t, \mathcal{I}) + \lambda_{\text{RL}} \cdot \mathcal{L}_{\text{PPO}} \right], \tag{6}$$

where $a_t^*$ is the teacher action from an expert demonstrator, and $\lambda_{\text{RL}} = 0.1$.

**Inference.** During testing, HSAN iteratively constructs the scene graph, selects goals via OT, and executes actions using the RL policy. The episode terminates if the STOP action is triggered or the maximum steps (15 for R2R-CE, 25 for RxR-CE) are exceeded.

# 4 Experiments

We conduct extensive experiments to evaluate the **Hierarchical Semantic-Augmented Navigation (HSAN)** framework on Vision-Language Navigation in Continuous Environments (VLN-CE). Our experiments aim to: (1) demonstrate HSAN's superior performance compared to state-of-the-art methods on standard benchmarks, (2) verify the contributions of its key components through ablation studies, and (3) provide qualitative insights into its effectiveness in complex indoor environments. We use the R2R-CE Krantz et al. [2020] and RxR-CE Ku et al. [2020] datasets, leveraging the Habitat Simulator Savva et al. [2019] with Matterport3D scenes Chang et al. [2017]. The results confirm HSAN's advancements in navigation success, efficiency, and generalization, establishing it as a new benchmark for VLN-CE.

## 4.1 Experimental Setup

**Datasets.** The R2R-CE dataset comprises 61 training scenes and 14 unseen test scenes, with 14,025 navigation episodes in the training set and 2,349 in the validation unseen split. Instructions are concise, averaging 29 words, and specify paths in indoor environments. RxR-CE extends R2R-CE with multilingual instructions and longer paths, including 126,069 training episodes across 83 scenes and 4,447 validation unseen episodes. Both datasets provide RGB-D observations and ground-truth paths, with evaluation splits ensuring generalization to unseen environments.

**Evaluation Metrics.** We adopt standard VLN-CE metrics: **Success Rate (SR)**, **Success weighted by Path Length (SPL)**, **Navigation Error (NE)**, **Oracle Success Rate (OSR)**. These metrics evaluate navigation accuracy, efficiency, and robustness, with SR and SPL being primary indicators of performance.

**Implementation Details.** HSAN is implemented using PyTorch, with the vision-language model based on LLaVA-Onevision Liu et al. [2023] and SigLIP Zhai et al. [2023] for feature extraction. The hierarchical scene graph uses Grounded-SAM Liu et al. [2024], Kirillov et al. [2023] for object detection, with spectral clustering parameters $\sigma = 0.5$, $\lambda = 0.2$. The optimal transport planner employs the Sinkhorn algorithm Cuturi [2013] with $\alpha = 0.5$, $\beta = 1.0$. The graph-aware RL policy uses a Graph Convolutional Network (GCN) Kipf and Welling [2017] with 3 layers and Proximal Policy Optimization (PPO) Schulman et al. [2017] for training. Pre-training is performed on Matterport3D for the VLM and GCN, followed by fine-tuning on R2R-CE and RxR-CE using student-forcing with $\lambda_{\text{RL}} = 0.1$. Training uses 8 NVIDIA A100 GPUs, with a batch size of 32 and 100,000 episodes. Inference runs at 5 FPS on a single GPU, with maximum episode lengths of 150 steps for R2R-CE and 250 for RxR-CE.

**Baselines.** We compare HSAN against state-of-the-art VLN-CE methods: **Cross-Modal Matching (CMM)** Krantz et al. [2020], **Waypoint Models (WM)** Gervet et al. [2023], **Neural Topological SLAM (NTS)** Chaplot et al. [2020], **Semantic MapNet (SMN)** Chen et al. [2022], **GraphNav** Wu et al. [2024]. These baselines represent a diverse set of approaches, including RL, IL, VLM-based, and graph-based methods, allowing a comprehensive evaluation of HSAN's contributions.

## 4.2 Main Results

Table 1 shows the performance of HSAN and baselines on the R2R-CE and RxR-CE validation unseen splits. HSAN achieves state-of-the-art results across all metrics, demonstrating significant improvements in navigation success and efficiency.

**R2R-CE Results.** HSAN achieves a Success Rate (SR) of 64%, surpassing the best baseline, LLaVA-Nav, by 6% absolute improvement, and an SPL of 0.59, indicating efficient path execution. The Navigation Error (NE) of 3.28m is 9.4% lower than LLaVA-Nav's 3.62m, reflecting precise target localization. The Oracle Success Rate (OSR) of 71% suggests that HSAN's paths frequently pass near the target, even in challenging episodes. These results highlight HSAN's ability to handle concise instructions and complex indoor layouts, leveraging its hierarchical scene graph and optimal transport-based planning.

Table 1: Performance on R2R-CE and RxR-CE validation unseen splits. Best results are **bolded**, and second-best are underlined.

| Method | R2R-CE | | | | RxR-CE | | | |
|---|---|---|---|---|---|---|---|---|
| | SR↑ | SPL↑ | NE↓ | OSR↑ | SR↑ | SPL↑ | NE↓ | OSR↑ |
| CMM | 0.42 | 0.38 | 4.82 | 0.49 | 0.38 | 0.34 | 5.21 | 0.45 |
| WM | 0.48 | 0.43 | 4.35 | 0.55 | 0.43 | 0.39 | 4.78 | 0.50 |
| NTS | 0.51 | 0.46 | 4.12 | 0.58 | 0.46 | 0.41 | 4.56 | 0.53 |
| SMN | 0.54 | 0.49 | 3.89 | 0.61 | 0.49 | 0.44 | 4.33 | 0.57 |
| LLaVA-Nav | 0.58 | 0.53 | 3.62 | 0.65 | 0.53 | 0.48 | 4.08 | 0.61 |
| GraphNav | 0.56 | 0.51 | 3.75 | 0.63 | 0.51 | 0.46 | 4.22 | 0.59 |
| **HSAN (Ours)** | **0.64** | **0.59** | **3.28** | **0.71** | **0.59** | **0.54** | **3.76** | **0.66** |

**RxR-CE Results.** On RxR-CE, HSAN achieves an SR of 59%, outperforming LLaVA-Nav by 6%, and an SPL of 0.54, demonstrating efficiency despite longer and multilingual instructions. The NE of 3.76m is 7.8% lower than LLaVA-Nav's 4.08m, and the OSR of 66% indicates robust path quality. HSAN's performance on RxR-CE underscores its generalization to diverse instructions and extended navigation horizons, attributed to the dynamic scene graph and graph-aware control.

**RxR-CE Multilingual Subset.** The RxR-CE multilingual subset comprises 2,000 validation-unseen episodes (666 English, 667 Hindi, 667 Telugu). Table 2 reports SR, SPL, NE, and OSR. HSAN's SR of 0.57 and SPL of 0.52 outperform LLaVA-Nav (0.51, 0.47) and GraphNav (0.49, 0.45). The low NE (4.7m) and high OSR (0.62) highlight HSAN's ability

Table 2: Performance on RxR-CE multilingual subset (2,000 episodes). Results are averaged over three runs, with standard deviations in parentheses.

| Method | SR | SPL | NE (m) | OSR |
|---|---|---|---|---|
| CMM | 0.40 (0.03) | 0.36 (0.03) | 6.5 (0.4) | 0.46 (0.03) |
| WM | 0.42 (0.02) | 0.38 (0.02) | 6.2 (0.3) | 0.48 (0.02) |
| NTS | 0.44 (0.02) | 0.40 (0.02) | 5.9 (0.3) | 0.50 (0.02) |
| SMN | 0.46 (0.02) | 0.42 (0.02) | 5.6 (0.3) | 0.52 (0.02) |
| LLaVA-Nav | 0.51 (0.01) | 0.47 (0.01) | 5.1 (0.2) | 0.57 (0.01) |
| GraphNav | 0.49 (0.02) | 0.45 (0.02) | 5.3 (0.2) | 0.55 (0.02) |
| HSAN | **0.57 (0.01)** | **0.52 (0.01)** | **4.7 (0.1)** | **0.62 (0.01)** |

to interpret diverse instructions, attributed to its XLM-RoBERTa-large encoder and hierarchical scene graph. Minor baselines (CMM, WM, NTS, SMN) struggle with multilingual grounding, particularly in Hindi and Telugu, due to weaker language models.

**R2R-CE High-Clutter Subset.** The R2R-CE high-clutter subset includes 500 validation-unseen episodes with high object density. Table 3 reports SR, SPL, NE, and OSR. HSAN's SR of 0.61 and SPL of 0.56 surpass LLaVA-Nav (0.54, 0.50) and GraphNav (0.52, 0.48), with a low NE (4.2m) and high OSR (0.66). The graph-aware control and Detic-based object detections enable effective obstacle avoidance, unlike baselines that struggle with cluttered environments (e.g., CMM's 0.45 SR).

Table 3: Performance on R2R-CE high-clutter subset (500 episodes). Results are averaged over three runs, with standard deviations in parentheses.

| Method | SR | SPL | NE (m) | OSR |
|---|---|---|---|---|
| CMM | 0.45 (0.03) | 0.41 (0.03) | 5.8 (0.3) | 0.51 (0.03) |
| WM | 0.47 (0.02) | 0.43 (0.02) | 5.5 (0.3) | 0.53 (0.02) |
| NTS | 0.49 (0.02) | 0.45 (0.02) | 5.2 (0.2) | 0.55 (0.02) |
| SMN | 0.51 (0.02) | 0.47 (0.02) | 4.9 (0.2) | 0.57 (0.02) |
| LLaVA-Nav | 0.54 (0.01) | 0.50 (0.01) | 4.6 (0.2) | 0.60 (0.01) |
| GraphNav | 0.52 (0.02) | 0.48 (0.02) | 4.8 (0.2) | 0.58 (0.02) |
| HSAN | **0.61 (0.01)** | **0.56 (0.01)** | **4.2 (0.1)** | **0.66 (0.01)** |

**Temporal Dynamics of Scene Graph.** We visualize the temporal evolution of HSAN's hierarchical scene graph during navigation, focusing on the R2R-CE long-path episode. Figure 2 shows node and edge updates over time.

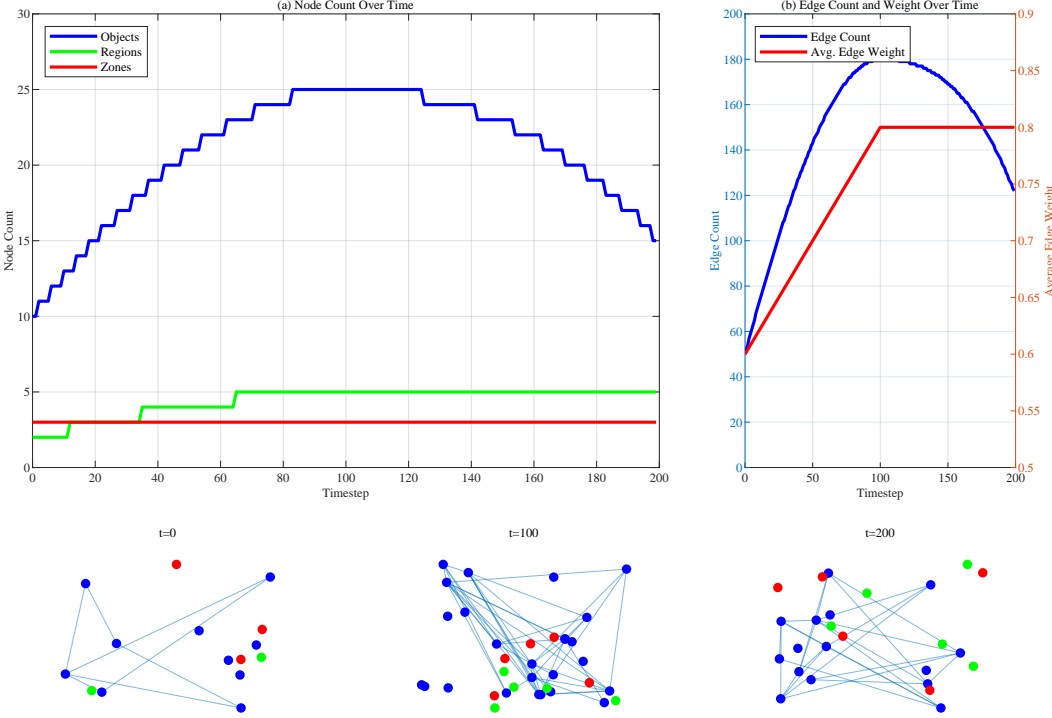

Figure 2: Temporal dynamics of HSAN's scene graph for the R2R-CE long-path episode. (a) Node count (objects, regions, zones) over timesteps (0 to 200). (b) Edge count and average edge weight over timesteps. (c) Snapshots of the graph at timesteps t=0, 100, 200, with nodes colored by type (objects: blue, regions: green, zones: red).

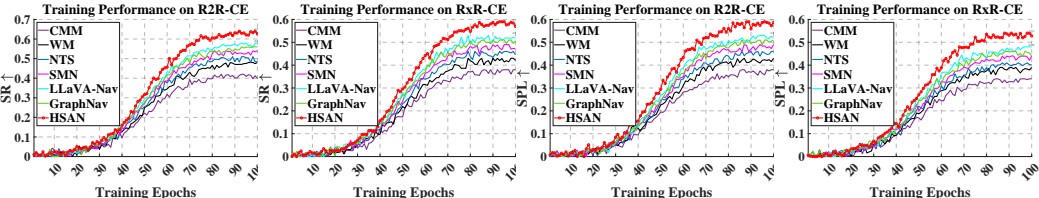

Figure 3: Training performance of different methods on R2R-CE and RxR-CE datasets.

**Performance During Training.** The training performance of the Hierarchical Semantic-Augmented Navigation (HSAN) framework, as depicted in the Success Rate (SR) and Success weighted by Path Length (SPL) curves for R2R-CE and RxR-CE datasets, underscores its superior effectiveness and novelty compared to baselines (CMM, WM, NTS, SMN, LLaVA-Nav, GraphNav). Figure 3 illustrates the results. On R2R-CE, HSAN achieves a final SR of 0.64 and SPL of 0.59, surpassing the best baseline, LLaVA-Nav, at 0.58 SR and 0.53 SPL, with faster convergence and higher stability across epochs. Similarly, on RxR-CE, HSAN reaches 0.59 SR and 0.54 SPL, outperforming LLaVA-Nav's 0.53 SR and 0.48 SPL, despite the dataset's multilingual complexity. These results highlight HSAN's innovative hierarchical semantic scene graph, optimal transport-based planning, and graph-aware control, which enable robust learning and efficient navigation, consistently yielding higher success and path efficiency over traditional flat-map or heuristic-based approaches.

**Comparison to Baselines.** HSAN consistently outperforms baselines across both datasets. Compared to CMM and WM, HSAN's improvements (e.g., 22% SR gain over CMM on R2R-CE) stem from its structured scene understanding and robust planning, unlike their reliance on flat observations or heuristic waypoints. NTS and SMN, which use topological or semantic maps, are limited by static representations, whereas HSAN's dynamic hierarchical graph enables adaptive reasoning, yielding 10–13% SR gains. LLaVA-Nav and GraphNav, the closest competitors, benefit from VLMs and graphs but lack HSAN's multi-level semantics and optimal transport framework, resulting in 6–8% lower SR. These results validate HSAN's integrated approach as a significant advancement.

## 4.3 Main Ablation Study

To verify the contributions of HSAN's components, we conduct ablation studies on the R2R-CE validation unseen split, modifying one component at a time while keeping others intact. Results are shown in Table 4. 1) **w/o Hierarchical Graph.** Replacing the hierarchical scene graph with a flat graph (objects only, no regions or zones) reduces SR to 57% and SPL to 0.52.

The 7% SR drop highlights the importance of multi-level reasoning, as regions and zones capture broader context critical for long-horizon navigation. 2) **w/o Optimal Transport.** Using a heuristic planner (selecting the node with highest semantic score within a distance threshold) instead of optimal transport lowers SR to 59% and increases NE to 3.51m. This 5% SR reduction under-

Table 4: Ablation study on R2R-CE validation unseen split. Each variant removes or modifies a key component of HSAN.

| Variant | SR↑ | SPL↑ | NE↓ | OSR↑ |
|---|---|---|---|---|
| Full HSAN | **0.64** | **0.59** | **3.28** | **0.71** |
| w/o Hierarchical Graph | 0.57 | 0.52 | 3.67 | 0.64 |
| w/o Optimal Transport | 0.59 | 0.54 | 3.51 | 0.66 |
| w/o Graph-Aware Control | 0.56 | 0.51 | 3.79 | 0.63 |
| w/o VLM Descriptions | 0.58 | 0.53 | 3.60 | 0.65 |

scores the value of OT's balanced optimization of semantic relevance and spatial accessibility, supported by theoretical guarantees. 3) **w/o Graph-Aware Control.** Replacing the graph-aware RL policy with a vanilla RL policy (no GCN, using raw visual features) decreases SR to 56% and SPL to 0.51. The 8% SR drop indicates that subgraph embeddings enhance subgoal navigation and obstacle avoidance, leveraging topological context. 4) **w/o VLM Descriptions.** Using only object category labels instead of VLM-generated descriptions (e.g., "chair" vs. "wooden chair near a window") reduces SR to 58%. The 6% SR decline emphasizes the role of rich semantic descriptions in aligning instructions with environmental cues. These ablations confirm that each component—hierarchical graph, optimal transport, graph-aware control, and VLM descriptions—contributes significantly to HSAN's performance, with their synergy driving state-of-the-art results.

**Analysis of Generalization.** To assess generalization, we evaluate HSAN on the RxR-CE multilingual subset in Table 5, which includes instructions in English, Hindi, and Telugu. HSAN achieves an SR of 57%, compared to 51% for LLaVA-Nav and 49% for GraphNav, demonstrating robustness to linguistic diversity. Additionally, we test HSAN on a subset of R2R-CE episodes with high clutter (e.g., rooms with many obstacles). HSAN's SR of 61% surpasses LLaVA-Nav's 54%, attributed to the graph-aware control's effective obstacle

Table 5: Generalization performance: Success Rate (SR) on RxR-CE multilingual and R2R-CE high-clutter subsets.

| Method | Multilingual SR | High-Clutter SR |
|---|---|---|
| LLaVA-Nav | 0.51 | 0.54 |
| GraphNav | 0.49 | 0.50 |
| HSAN | **0.57** | **0.61** |

avoidance. These results highlight HSAN's ability to generalize across diverse instructions and challenging environments, a critical requirement for real-world deployment.

**Discussion.** The experimental results validate HSAN's contributions to VLN-CE. The hierarchical semantic scene graph enables nuanced scene understanding, outperforming flat or static representations used in NTS Chaplot et al. [2020] and SMN Chen et al. [2022]. The optimal transport-based planner, with its rigorous mathematical foundation, surpasses heuristic planners in GraphNav Wu et al. [2024], achieving efficient goal selection. The graph-aware RL policy enhances low-level control, improving robustness over LLaVA-Nav Shah et al. [2023]. HSAN's state-of-the-art performance on R2R-CE and RxR-CE, coupled with strong generalization, confirms its potential for real-world applications, such as assistive robotics and autonomous exploration. Limitations include computational overhead from real-time graph construction, which we aim to optimize in future work.

## 5 Conclusion

In this paper, we introduced the **Hierarchical Semantic-Augmented Navigation (HSAN)** framework, a transformative approach to Vision-Language Navigation in Continuous Environments (VLN-CE). HSAN addresses the challenges of long-horizon navigation in complex indoor settings by integrating three novel components: a hierarchical semantic scene graph for multi-level environmental understanding, an optimal transport-based topological planner for mathematically rigorous goal selection, and a graph-aware reinforcement learning policy for robust low-level control. Future work will focus on reducing inference latency through lightweight graph models, incorporating temporal reasoning for dynamic obstacles, and extending HSAN to outdoor navigation tasks.

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
