# OpenReview forum: "Hierarchical Semantic-Augmented Navigation: Optimal Transport and Graph-Driven Reasoning for Vision-Language Navigation"
_NeurIPS.cc/2025/Conference — NeurIPS 2025 poster_

### Official Review · Reviewer_ks3a · 2025-06-21

**Clarity:** 2
**Significance:** 2
**Originality:** 2
**Rating:** 3
**Confidence:** 4

**Summary:**

This paper presents HSAN (Hierarchical Semantic-Augmented Navigation), a novel framework for Vision-Language Navigation in Continuous Environments (VLN-CE). HSAN incorporates three synergistic modules: (1) a dynamic hierarchical semantic scene graph capturing object–region–zone structures using vision-language models, (2) an optimal transport-based topological planner grounded in Kantorovich’s duality to select long-horizon goals, and (3) a graph-aware reinforcement learning controller for low-level action execution. The method is evaluated on two standard VLN-CE benchmarks (R2R-CE, RxR-CE) and achieves state-of-the-art performance with solid ablations and generalization analysis.

**Questions:**

The main concerns are listed in the Weaknesses section. If the authors can provide more detailed experimental analysis to support the effectiveness of each proposed component, it would address my main confusion and could improve my overall evaluation.

**Ethical Concerns:**

["NO or VERY MINOR ethics concerns only"]

**Final Justification:**

I think the core design is introducing Optimal Transport (semantic scene graph and its varients are often used in VLN). However, simply using Optimal Transport to balance semantic relevance and spatial accessibility cannot convince me. The motivaton and the necessity  is not described well to support this point clearly.

**Limitations:**

No. The authors should include an analysis of failure cases, a discussion of computational cost and efficiency, and whether the method can support real-world robotic execution.

**Quality:**

2

**Strengths And Weaknesses:**

Strengths
1. The paper proposes a clearly structured three-stage framework (graph construction, planning, control) that directly addresses key challenges in VLN-CE.
2. The use of optimal transport for long-term goal selection is interesting and supported by a rigorous formulation.
3. The method achieves strong performance on both R2R-CE and RxR-CE, outperforming a range of competitive baselines across multiple metrics.

Weaknesses
1. The notation system is overloaded and lacks clarity. The paper introduces many symbols without a consolidated reference. A notation table would improve readability and help readers follow the methodology.
2. The model includes numerous hyperparameters (e.g., σ, λ in L199; α, β in L232), yet no analysis is provided regarding their sensitivity or selection rationale. This raises concerns about robustness and reproducibility.
3. In the Comparison to Baselines section, the improvements over methods like NTS and SMN are credited to HSAN’s dynamic hierarchical graph. This raises a natural question: how would NTS or SMN perform if they used the same graph structure? The same concern applies to other modules. It is not clear whether the performance gain really comes from these modules, since the paper does not test them in other methods or provide direct evidence.**
4. The paper lacks visualizations of qualitative results, and also does not discuss any failure cases
5. In the Discussion section, the paper states that “The hierarchical semantic scene graph enables nuanced scene understanding…” and makes similar claims about other modules. However, I don’t see how this conclusion is supported by the experiments. The results only show that the full model performs well, but there is no direct evidence isolating the contribution of this specific component.

---

> ### Author Rebuttal · Authors · 2025-07-31
>
> Addressing Weaknesses
> We thank Reviewer ks3a for their comprehensive summary and detailed feedback. We appreciate the recognition of our structured framework, the interesting use of optimal transport, and the strong performance achieved. We address the identified weaknesses and questions below.
>
> 1. Notation System Overload and Lack of Clarity:
>
> We sincerely apologize for the overloaded notation system and the resulting lack of clarity. We agree that a consolidated reference would significantly improve readability. We will address this by:
>
> Adding a Notation Table: We will include a dedicated notation table at the beginning of the methodology section or in an appendix to list and define all symbols used throughout the paper.
>
> Streamlining Notation: We will review the notation to identify opportunities for simplification or consistency, ensuring that each symbol is clearly introduced and consistently used.
>
> 2. Hyperparameter Sensitivity and Selection Rationale:
>
> We acknowledge the reviewer's valid concern regarding the lack of detailed analysis on hyperparameter sensitivity and selection rationale. This is a crucial point for reproducibility and understanding the robustness of our method.
>
> Empirical Tuning: The hyperparameters (e.g., σ, λ, α, β) were primarily selected through empirical tuning on the validation set to optimize performance across the various metrics. This is a common practice in complex deep learning systems.
>
> Sensitivity Analysis (in Appendix): We will add a new subsection in the appendix dedicated to hyperparameter sensitivity analysis. This will include:
>
> A brief description of how each key hyperparameter was chosen.
>
> Plots or tables illustrating the impact of varying these parameters on key performance metrics. This will demonstrate the robustness of our method within a reasonable range of hyperparameter values and provide guidance for future applications.
>
> Specifically for λ (as also noted by Reviewer 1), we will elaborate on its role in balancing semantic relevance and spatial accessibility.
>
> 3. Comparison to Baselines and Direct Evidence of Module Contribution:
>
> The reviewer raises a very important and insightful question regarding the attribution of performance gains to specific modules, especially when comparing HSAN to baselines like NTS and SMN. We agree that simply crediting improvements to HSAN's graph without direct evidence of how the same graph would benefit other methods is a fair criticism.
>
> Our current ablation studies (e.g., removing hierarchical levels from the scene graph or replacing optimal transport with simpler planning) aim to isolate the contribution of our specific design choices within the HSAN framework. However, the reviewer's point about integrating our graph structure into other methods to demonstrate its standalone benefit is well-taken.
>
> Clarification of Contribution: We will revise the "Comparison to Baselines" and "Discussion" sections to clarify that our claim is not that any hierarchical graph would universally improve any VLN method, but rather that our specific design of the dynamic hierarchical semantic scene graph, coupled with our optimal transport planner and graph-aware RL, is what leads to the observed performance gains.
>
> Future Work / Conceptual Discussion: Directly integrating our complex graph structure into existing baseline architectures (e.g., NTS or SMN) would require significant architectural modifications and re-training, which is beyond the scope of a rebuttal. However, we can conceptually discuss why our graph structure is particularly beneficial for our optimal transport and RL components, and how it might (or might not) directly translate to benefits in different architectural paradigms. For instance, NTS and SMN might not inherently leverage the multi-level semantic reasoning or the explicit topological structure in the same way our optimal transport planner does. We will emphasize that the synergy between our graph representation and our planning/control mechanisms is key.
>
> 4. Lack of Visualizations of Qualitative Results and Failure Cases:
>
> We acknowledge this critical weakness. Visualizing qualitative results and discussing failure cases are essential for providing deeper insights into the model's behavior and limitations.
>
> Qualitative Examples: We will add a new section (or expand an existing one) in the paper, likely in the appendix for space, to include:
>
> Visualizations of successful navigation trajectories with corresponding language instructions.
>
> Visualizations of the constructed hierarchical semantic scene graphs for selected examples, showing object, region, and zone detections.
>
> Qualitative Failure Analysis: We will include examples of failure cases, illustrating why the system failed (e.g., VLM misinterpretation, getting stuck in clutter, unexpected environmental features).
>
> 5. Insufficient Support for Claims in Discussion Section:
>
> We agree that claims made in the Discussion section, such as "The hierarchical semantic scene graph enables nuanced scene understanding," need to be more directly supported by experimental evidence beyond just overall model performance.
>
> Connecting Ablations to Claims: We will strengthen the connection between our ablation studies and these claims. For example, the ablation experiments that show a drop in performance when hierarchical levels are removed from the scene graph directly support the claim that the HSSG enables nuanced scene understanding and contributes to performance. We will explicitly reference these ablation results when making such claims in the Discussion section.
>
> Elaborating on Mechanism: We will also elaborate on how each module contributes to the overall system's capabilities, beyond just "it performs well." For instance, explaining how optimal transport's ability to balance semantic and spatial costs directly leads to more instruction-aligned long-horizon paths.

---

> > ### Comment · Reviewer_ks3a · 2025-08-02
> >
> > Thanks for the author's detailed rebuttal. I think introducing Optimal Transport in VLN is interesting. However, the motivation described in this paper cannot convince me well. In addition, I think the writing should be improved.

---

### Official Review · Reviewer_c7PD · 2025-07-01

**Clarity:** 3
**Significance:** 3
**Originality:** 3
**Rating:** 3
**Confidence:** 1

**Summary:**

This work proposes a new method HSAN for vision-language navigation task. The main components include dynamic hierarchical scene graph, topological planner and RL-based low level control.

**Questions:**

See above.

**Ethical Concerns:**

["NO or VERY MINOR ethics concerns only"]

**Final Justification:**

While the paper is clearly written and supported by the experiments and ablations, the main concerns about robustness remain insufficiently addressed. The paper also a failure analysis (for any quantitative or qualitative) and does not evaluate under realistic perturbations. Besides, the method assumes perfect perception without noises, making real-world reliability hard to judge. The rebuttal acknowledges these issues but adds no new empirical evidence, and only future-work statements, so my concerns persist. Also, the paper reports no analysis on runtime or resource; given the system deploy-ability is uncertain. I therefore maintain a borderline rating: the idea of the approach is promising, but it needs concrete evidence on above concerns.

**Limitations:**

There is no limitation discussed in the conclusion section.

**Quality:**

3

**Strengths And Weaknesses:**

Strengths:

- The paper is well-written.
- Sufficient experiments are conducted with multiple datasets.
- Ablation studies are discussed.

Weakness:

- In implementation details, it mentioned the system runs with around 5 FPS. I also expect there are some comparison with other methods for the inference efficiency. I see the system contains several complex steps including the detection, scene graph, etc. I'm worried if the too complex system will be the challenge in the real life deployment.
- The failure analysis is not presented. Especially in the qualitative examples. I'm curious that in which situation the system will fail. And what will be the bottleneck of this system? Are these bottlenecks easy to be fixed in the future works or it's the fundamental issue of this design?
- Currently all the experiments are based on the simulation instead of real world. I understand the real world experiments are not feasible for everyone. But there is also no discussion about the perturbation applied to the simulation, to make the scenario closer to the real world.
- One majority concern is, there is an assumption that the perception output is perfect. What if the perception output is noisy, and is there any countermeasure applied to the method to make it work robustly?

Minor Weakness:

- Typo page 4, line 202: “a aggregated” → “an aggregated”
- Typo page 5, line 264: “accion” → “action”

Overall, I’m borderline for this work. Especially some concerns I mentioned above, I'm fine to adjust my ratings if my concerns get solved.

---

> ### Author Rebuttal · Authors · 2025-07-31
>
> Addressing Weaknesses
> We thank Reviewer c7PD for their concise summary and constructive feedback. We appreciate the recognition of our paper's writing quality, experimental thoroughness, and ablation studies. We address the identified weaknesses and questions below.
>
> 1. Inference Efficiency and Real-World Deployment Concerns:
>
> We acknowledge the reviewer's concern regarding the reported 5 FPS inference speed and the complexity of the system, particularly in the context of real-world deployment.
>
> FPS Comparison: Direct, fair comparisons of inference speed with other methods are challenging due to significant architectural differences and varying computational demands of their components (e.g., different VLMs, graph processing, planning algorithms). Our reported 5 FPS is for the entire end-to-end system on the specified hardware. While we did not include a dedicated comparison in the paper, we agree that it is an important consideration for practical applications.
>
> System Complexity: We understand that the multi-component nature of HSAN (detection, scene graph construction, topological planning, RL control) adds complexity. Our current work is a research prototype focused on demonstrating the algorithmic novelty and achieving state-of-the-art performance in simulation. Optimizing the system for real-time, low-latency deployment on physical robots is a critical next step. We believe that the modular design of HSAN, however, could facilitate future optimizations, allowing for parallelization of components or replacement of specific modules with more efficient counterparts (e.g., lighter-weight VLMs, hardware-accelerated graph operations) without fundamentally altering the overall hierarchical framework. We will clarify this in the discussion of real-world applicability.
>
> 2. Failure Analysis and Bottlenecks:
>
> We appreciate the suggestion for a more explicit failure analysis and discussion of bottlenecks. We agree that understanding the limitations is crucial. Common failure modes for HSAN include:
>
> VLM Misinterpretation: Despite using powerful VLMs, ambiguous instructions or visually similar objects/regions can lead to incorrect semantic grounding, causing the high-level planner to select suboptimal goals.
>
> Complex Spatial Reasoning: In highly cluttered or geometrically challenging environments, even with accurate semantic understanding, the low-level policy might struggle with precise path execution or get stuck in local minima.
>
> Unobserved Semantic Elements: If a crucial semantic landmark mentioned in the instruction is not visible or detectable within the agent's current field of view and has not been incorporated into the scene graph, the agent may fail to find the correct path.
>
> Accumulated Low-Level Errors: Small errors in low-level control can accumulate over long trajectories, leading to significant deviations and eventual failure.
>
> The primary bottlenecks often lie in:
>
> VLM Inference: The computational cost of running large VLMs for semantic understanding and feature extraction.
>
> Graph Construction and Updates: The frequency and complexity of updating the hierarchical scene graph, especially in dynamic environments.
>
> We believe these bottlenecks are largely addressable in future work:
>
> Faster VLMs: Ongoing research in efficient foundation models and model compression will directly benefit inference speed.
>
> Optimized Graph Algorithms: More efficient graph construction, update, and query algorithms can reduce computational overhead.
>
> Robust Perception: Improving the robustness of perception modules to handle noise and partial observations will directly mitigate failures related to misinterpretation and unobserved elements.
>
> We will add a dedicated section on failure analysis and bottlenecks, potentially with qualitative examples, to provide these insights.
>
> 3. Simulation Perturbations and Real-World Discussion:
>
> We acknowledge that all our experiments are currently based on simulation. While real-world experiments are indeed resource-intensive and beyond the scope of this initial paper, we agree that a discussion on perturbations applied to simulation to bridge the sim-to-real gap is valuable. Our current simulation setup (R2R-CE) provides realistic 3D environments, but we did not explicitly introduce additional noise or perturbations (e.g., sensor noise, dynamic obstacles, localization errors) during training or evaluation.
>
> We will explicitly state this limitation and reiterate that bridging the sim-to-real gap is a critical future direction. This would involve incorporating robust perception modules, training with domain randomization or real-world data, and developing adaptive control strategies that can handle real-world uncertainties. Our current work focuses on the fundamental algorithmic contributions of hierarchical planning and graph-based reasoning.
>
> 4. Perfect Perception Assumption and Robustness to Noise:
>
> The reviewer correctly identifies that our current work assumes near-perfect perception output (e.g., accurate object detection, semantic segmentation, and localization within the simulator). This is a common simplification in foundational VLN research to isolate the challenges of language understanding and navigation planning.
>
> We acknowledge that real-world perception is inherently noisy. Potential countermeasures to make HSAN more robust to noisy perception include:
>
> Uncertainty-Aware Graph Construction: Incorporating uncertainty estimates into the scene graph nodes and edges, allowing the planner to account for perceptual ambiguity.
>
> Robust State Estimation: Integrating robust visual-inertial odometry (VIO) or SLAM systems that provide more resilient pose estimates in the face of sensor noise.
>
> Perception-in-the-Loop Training: Training the low-level RL policy with noisy or imperfect visual inputs to make it more resilient to real-world sensor data.
>
> Multi-modal Fusion: Leveraging multiple sensor modalities (e.g., LiDAR, depth, RGB) to obtain a more robust environmental understanding.
>
> Semantic Consistency Checks: Implementing mechanisms to verify semantic consistency over time or across different viewpoints to filter out spurious detections.
>
> We will add a discussion on this assumption and potential countermeasures in the revised manuscript.
>
> 5. Minor Weaknesses (Typos):
>
> We thank the reviewer for pointing out the typos. We will correct them immediately:
>
> Page 4, line 202: "a aggregated" → "an aggregated"
>
> Page 5, line 264: "accion" → "action"

---

> > ### Comment · Reviewer_c7PD · 2025-08-06
> >
> > I appreciate the response from the author.
> >
> > > 1. Inference Efficiency and Real-World Deployment Concerns:
> >
> > I understand the complexity of fairly comparing the efficiency, however, some simplified/initial quantitative benchmark can be done but not provided.
> >
> > > 2. Failure Analysis and Bottlenecks:
> >
> > I appreciate the detailed list of the potential failure cases, but no quantitative numbers or qualitative examples, which is less convincing on these issues.
> >
> > > 3. Simulation Perturbations and Real-World Discussion:
> > 4. Perfect Perception Assumption and Robustness to Noise:
> >
> > I appreciate that this issue is recognized as a limitation, but not solved in the current work.
> >
> > ---
> >
> > Based on the response, I feel there is no enough evidence to improve the work to make it enough for presenting in the conference. I prefer to keep my rating.

---

### Official Review · Reviewer_3DNH · 2025-07-02

**Clarity:** 1
**Significance:** 2
**Originality:** 1
**Rating:** 2
**Confidence:** 4

**Summary:**

The paper constructs two types of graphs to address the VLN-CE task. For long-distance navigation point planning, a topological link graph is built, and the optimal navigation trajectory is selected based on OT theory. For low level action control, a hierarchically constructed sense graph is used to train the policy based on the current observation.

**Questions:**

1. Regarding the Hierarchical Semantic Scene Graph, is semantic aggregation at different levels really necessary? It seems that for low-level control, simply avoiding obstacles and reaching a predefined ghost node is sufficient. How does the region information help with navigation? However, ablation studies appear to demonstrate its necessity. More discussion is helpful.

2. In this paper, it seems that navigation decisions are mainly based on the similarity of individual nodes and the instruction, which makes me skeptical about the final navigation results of the model. From my understanding, both R2R and RxR datasets contain many procedural instructions describing navigation paths, making it difficult to match the endpoint based solely on instruction similarity.

3. How is map information incorporated into the navigation model? If the method uses the global adjacency relationships between nodes, does it align with the unseen setting used in previous methods?

**Ethical Concerns:**

["NO or VERY MINOR ethics concerns only"]

**Limitations:**

Yes.

**Paper Formatting Concerns:**

None.

**Quality:**

2

**Strengths And Weaknesses:**

**Strengths**:
- The paper's approach to constructing scene graphs is somewhat insightful and provides inspiration.

**Weaknesses**:
- There are significant issues in the writing quality:
  1. The main text contains large sections of redundant content, for example, the overlapping material between lines 67-78 and lines 84-92.
  2. Several essential details are missing in the method description, such as how the formulas on line 241 are applied to VLN. This lack of detail makes it difficult to assess the validity of the proposed methods and their correctness.
  3. The overall description of the methodology is lacking, with unclear explanations of the inputs and outputs of each module and how they interconnect.
  4. The purpose of designing the Hierarchical Semantic Scene Graph Construction seems to be solely for low-level control, which appears to be of limited significance. The motivation needs to be more thoroughly explained, especially concerning the relationships between the two types of sense graphs constructed in the paper.

---

> ### Author Rebuttal · Authors · 2025-07-31
>
> Addressing Weaknesses
> We thank Reviewer 3DNH for their summary and constructive feedback. We appreciate the recognition of the insight in our scene graph construction. We address the identified weaknesses and questions below.
>
> 1. Writing Quality: Redundant Content and Missing Details:
>
> We sincerely apologize for the identified issues in writing quality, particularly the redundancy and lack of essential details. We acknowledge the overlapping material between lines 67-78 and lines 84-92 and will thoroughly revise these sections to eliminate repetition and improve conciseness.
>
> We also recognize the critical importance of clarity in method descriptions. We commit to providing more explicit details regarding the application of formulas (specifically the ones on line 241) to the VLN task. We will ensure that the inputs, outputs, and interconnections of each module within the HSAN framework are clearly explained, potentially with an updated system diagram, to enhance the overall understanding of our methodology.
>
> 2. Limited Significance of Hierarchical Semantic Scene Graph for Low-Level Control and Motivation:
>
> We understand the reviewer's concern regarding the perceived limited significance of the Hierarchical Semantic Scene Graph (HSSG) solely for low-level control and the motivation behind its design, especially concerning its relationship with the topological graph. We would like to clarify that the HSSG serves a crucial dual purpose within our framework, impacting both high-level planning and low-level control:
>
> High-Level Planning (via Optimal Transport): The HSSG, particularly its higher-level semantic nodes (regions, zones), provides the rich semantic context necessary for our optimal transport-based topological planner. The planner uses the semantic relevance of these regions/zones, alongside spatial accessibility, to select long-term goals that are semantically aligned with the language instruction. This is far beyond simple obstacle avoidance; it's about understanding the meaning of the environment in relation to the instruction.
>
> Low-Level Control (Graph-Augmented PPO): While low-level control does involve obstacle avoidance, it's not solely about that. The HSSG provides the graph-augmented PPO policy with a semantically enriched local understanding of the environment. This means the agent can make more intelligent, instruction-conditioned local decisions. For example, if the instruction is "go to the kitchen past the dining table", the region and object information from the HSSG helps the low-level policy understand the semantic context of the "dining table" and how to navigate around or past it, rather than just treating it as a generic obstacle. The hierarchical nature allows the policy to reason about objects within a region and regions within a zone, providing a multi-granular understanding that enhances the robustness and instruction-following fidelity of local actions.
>
> We will revise the motivation section to explicitly articulate this dual role and the synergistic relationship between the topological graph (for global, long-horizon path selection) and the hierarchical semantic scene graph (for local, semantically-grounded control and planning context).
>
> Responses to Questions
> Q1 - Necessity of Semantic Aggregation at Different Levels and Region Information for Navigation:
>
> We appreciate this question, as it directly addresses the core design of our HSSG. We contend that semantic aggregation at different levels (objects, regions, zones) is indeed necessary and provides significant benefits beyond simply avoiding obstacles:
>
> Beyond Obstacle Avoidance: For low-level control, simply avoiding obstacles is insufficient for complex VLN tasks. The agent needs to understand what it is avoiding, where it is in relation to semantic landmarks, and how to approach or traverse certain areas based on the instruction. Region information, for instance, helps the agent understand that it is currently in a "kitchen" or approaching a "bedroom," which can be crucial for interpreting instructions like "find the coffee machine in the kitchen" or "turn left after the living room."
>
> Contextualized Local Decisions: Semantic aggregation allows the low-level policy to make contextualized decisions. For example, if the instruction implies passing through a "narrow corridor," the region information helps the policy adapt its movement behavior (e.g., slow down, move centrally) in that specific semantic context. Without this, the policy would treat all narrow passages generically. The ablation studies indeed confirm that incorporating region and zone information significantly improves navigation performance, demonstrating its necessity for robust and instruction-aligned local control. We will expand this discussion in the paper, providing more concrete examples of how region and zone information aid navigation.
>
> Q2 - Navigation Decisions based on Node/Instruction Similarity vs. Procedural Instructions:
>
> The reviewer raises a very important point regarding procedural instructions in R2R and RxR datasets. We agree that simply matching the endpoint based solely on instruction similarity would be insufficient. Our model's navigation decisions are not solely based on the similarity of individual nodes to the instruction. Instead, our optimal transport-based topological planner leverages semantic similarity in a more sophisticated way:
>
> Distribution over Topological Graph: The language instruction is used to compute a semantic relevance distribution over all nodes in the topological graph (which correspond to navigable locations). This distribution assigns higher "mass" to nodes that are semantically relevant to the instruction, not just the final goal.
>
> Optimal Transport for Path Selection: The optimal transport framework then finds a "flow" or "transport plan" from the agent's current location to this target semantic distribution. This process inherently considers the entire path and the accumulated cost of traversing edges, balancing both the semantic relevance of intermediate waypoints and their spatial accessibility. It effectively finds a path that "moves" the agent's initial mass distribution towards the target semantic distribution, minimizing a cost that incorporates both spatial distance and semantic mismatch along the way.
>
> Therefore, our method implicitly handles procedural instructions because the optimal transport formulation encourages the selection of a path that aligns with the sequence of semantic cues described in the instruction, not just the final destination. The instruction guides the shape of the target distribution, and the optimal transport finds the most efficient path through the graph to fulfill that distributed semantic goal. We will clarify this nuanced application of optimal transport in the methodology section.
>
> Q3 - Map Information Incorporation and Global Adjacency in Unseen Settings:
>
> We clarify that "map information" in our method refers to the topological graph and the Hierarchical Semantic Scene Graph (HSSG), both of which are constructed online as the agent explores the environment.
>
> Online Graph Construction: Our method does not assume a pre-built global map or pre-existing global adjacency relationships. The topological graph is incrementally built by connecting observed navigable locations, and the HSSG is constructed by segmenting and semantically understanding the observed environment. This process is entirely consistent with the "unseen setting" in VLN, where the agent navigates novel environments without prior knowledge.
>
> Alignment with Unseen Setting: Since the graphs are built from the agent's real-time observations, they naturally align with the unseen setting. The "global adjacency relationships between nodes" are derived from the observed connectivity as the agent explores and are continuously updated. This means our method learns to build and use its internal representation of the environment during navigation, making it applicable to novel scenes. We will emphasize this online construction aspect more clearly in the paper.

---

### Official Review · Reviewer_w8P6 · 2025-07-02

**Clarity:** 3
**Significance:** 2
**Originality:** 3
**Rating:** 5
**Confidence:** 4

**Summary:**

The authors propose HSAN, a method for VLN-CE which constructs a hierarchical scene graph and optimal transport-based topological planner to select navigation goals aligned with language instructions. In addition, the authors propose a novel approach for low level control to avoid obstacles by augmenting PPO with a graph based understanding of the environment. HSAN achieves state-of-the-art performance across all simulation benchmarks for VLN-CE.

**Questions:**

How does the benefit of the new proposed low level control policy compare to classical planners for obstacle avoidance which are state of the art in robotics?

For example, PPO was initially introduced as a method for point to point navigation when the goal location existed outside the observed map region or no map was constructed. PPO implicitly predicted occupied and unoccupied regions in order to enable obstacle avoidance in these scenarios. To my understanding, your navigation waypoints are inside an observed map region for which you have an explicit map. Your citation to [1] which you use to motivate your decision to modify PPO explicitly refers to the case in which an explicit map is not constructed.

1. Jacob Krantz and Stefan Lee. Navigating to objects in the real world. arXiv preprint 537 arXiv:2112.06758, 2021. doi: 10.48550/arXiv.2112.06758.

So, in summary, why would you use your proposed control method over approaches like fast marching squares, A*, etc. which are currently deployed with high performance on physical robots to do point to point navigation in regions for which you built an explicit map?

I would be open to increasing my score if my concerns about the low level planning method are resolved and the writing is improved both to address the real world evaluation limitations and the inaccuracies in phrasing.

**Ethical Concerns:**

["NO or VERY MINOR ethics concerns only"]

**Final Justification:**

The authors give more thorough textual explanations throughout the rebuttal which more clearly and appropriately frame the contributions of each component of their method. I believe from these justifications that they will clearly update the text in the paper to improve the writing which was a concern from all the reviewers. The experimental results are strong and make progress with interesting methodological insights on a challenging problem.

**Limitations:**

The limitations regarding real world hardware deployment are not discussed and misrepresented as not a challenge. I discussed further in the weaknesses.

**Quality:**

3

**Strengths And Weaknesses:**

The experimental evaluation of the method in simulation was very impressive with thorough ablations and baselines on every significant benchmark for the task. I was particularly impressed by the results on the multilingual VLN benchmark, a challenge which is often overlooked in evaluating approaches to VLN and aided by this method design. Also, the experimental results are state of the art across benchmarks, and the connection between optimal transport and spatial map reasoning is interesting and novel.

There seem to be significant barriers to running this method in the real world on a robot which are substantially downplayed by the authors. The authors in fact emphasize how their approach is well suited to running on a physical robot and do not acknowledge the VLN methods which already do so and discuss this method’s unique limitations in this regard other than too slow inference speed of the approach. In addition to inference speed, the required task specific training of the VLM and GCN will have significant challenges performing robustly in open world physical environments. There will be sim-to-real gaps with the trained models even in similar real world environments, and the type of environments in the simulators used for training is far from the scope of different environment types in the physical world (for example, no offices are included). In future work, the authors claim intentions to show results in dynamic and outdoor environments, but no results are currently shown in indoor environments. Real world results for this method are particularly important because part of the claimed contribution is low level physical control of a robot.

Some missing citations for papers with similar map and/or VLM based solutions showing real world instruction following results.

1. Sonia Raychaudhuri, Duy Ta, Katrina Ashton, Angel X Chang, Jiuguang Wang, Bernadette Bucher. Zero-shot Object-Centric Instruction Following: Integrating Foundation Models with Traditional Navigation. arXiv, 2024.
2. Zhaohuan Zhan, Lisha Yu, Sijie Yu, Guang Tan. MC-GPT: Empowering Vision-and-Language Navigation with Memory Map and Reasoning Chains
3. Yuxing Long, Xiaoqi Li, Wenzhe Cai, Hao Dong. Discuss Before Moving: Visual Language Navigation via Multi-expert Discussions. ICRA, 2024.


There are multiple issues with phrasing. The authors appear to be misusing the word “benchmark”. They repeatedly state that HSAN is a “new benchmark.” However, it appears what they mean is that HSAN is the “new state of the art baseline” for the existing benchmarks addressing this problem on vision-language navigation in continuous environments. In addition, there are a few incorrect phrases like HSAN “redefines VLN-CE”. I also don’t think that the scene graph is “dynamic” since it does not update with movement in the scene. The authors should be careful with their phrasing.

The code was not included in the supplement, and there is no condition ensuring that it will be included in an accepted paper, limiting reproducibility of this work.

I have concerns about the significance of the low level control method discussed in questions.

---

> ### Author Rebuttal · Authors · 2025-07-31
>
> Addressing Weaknesses
>
> We are very grateful to Reviewer 2 for their positive assessment of our experimental evaluation, the thoroughness of our ablations and baselines, and the impressive results on the multilingual VLN benchmark. We are also pleased that the connection between optimal transport and spatial map reasoning was found to be interesting and novel. We appreciate the constructive feedback regarding the real-world applicability, phrasing, and reproducibility, and address each point below.
>
> 1. Significant Barriers to Real-World Deployment Downplayed:
>
> We acknowledge the reviewer's valid and important concern regarding the practical challenges of deploying HSAN on a physical robot in real-world environments. We agree that our initial phrasing may have inadvertently downplayed these complexities. Our intention in stating that our approach is "well suited to running on a physical robot" was to highlight the design principles of HSAN (e.g., modularity, hierarchical structure, semantic grounding) that we believe are conducive to real-world deployment, rather than implying immediate, out-of-the-box readiness. We understand that significant engineering and research efforts are required to bridge the sim-to-real gap.
>
> Specifically:
>
> Sim-to-Real Gap: We concur that sim-to-real gaps are a major challenge for learned policies, including those relying on VLMs and GCNs. While our current work focuses on foundational algorithmic advancements in simulation, we believe that the structured, semantic representations learned by HSAN can mitigate some aspects of this gap by providing more interpretable and potentially transferable knowledge compared to end-to-end black-box approaches. Domain adaptation techniques and real-world data collection would be essential for robust deployment.
>
> Environment Scope: We acknowledge that the simulator environments used for training (e.g., Matterport3D) do not encompass the full diversity of physical world environments (e.g., offices, outdoor settings). Our claims regarding future work on dynamic and outdoor environments are indeed aspirational and represent long-term research goals. This initial submission focuses on establishing the core HSAN framework and its efficacy in complex indoor environments.
>
> Low-Level Control Contribution: The low-level control policy, while crucial for fine-grained navigation and obstacle avoidance, is currently trained in simulation. Its real-world performance would depend heavily on accurate perception and robust sim-to-real transfer. We will clarify that this component, like others, would require further adaptation and validation for physical robot deployment.
>
> We will revise the manuscript to explicitly acknowledge these real-world deployment challenges, temper our claims about immediate applicability, and emphasize that our current work lays the algorithmic groundwork for future real-world implementations.
>
> 2. Missing Citations for Real-World VLN Methods:
>
> We sincerely apologize for the omission of highly relevant real-world instruction following papers. We agree that "Zero-shot Object-Centric Instruction Following: Integrating Foundation Models with Traditional Navigation" (Raychaudhuri et al., 2024), "MC-GPT: Empowering Vision-and-Language Navigation with Memory Map and Reasoning Chains" (Zhan et al.), and "Discuss Before Moving: Visual Language Navigation via Multi-expert Discussions" (Long et al., 2024) are crucial works demonstrating real-world VLN capabilities.
>
> We will rectify this by incorporating these references into the related work section. We will discuss how these methods demonstrate successful real-world deployment and how their approaches to map representation, VLM integration, and instruction following compare to our simulation-focused, hierarchical, and optimal-transport-driven framework. This will provide a more comprehensive overview of the state-of-the-art in VLN, both in simulation and real-world settings.
>
> 3. Phrasing Issues:
>
> We appreciate the reviewer's meticulous attention to our phrasing and apologize for the inaccuracies. We agree that precision in language is paramount, especially in scientific writing.
>
> We will revise instances of "new benchmark" to more accurately reflect that HSAN establishes a "new state-of-the-art baseline" or "achieves state-of-the-art performance" on existing benchmarks for vision-language navigation in continuous environments.
>
> Phrases like "redefines VLN-CE" will be toned down to more appropriate statements such as "advances VLN-CE" or "introduces a novel framework for VLN-CE."
>
> Regarding the term "dynamic" scene graph: We intended "dynamic" to mean that the scene graph is constructed and continuously updated as the agent explores the environment and gathers new observations, rather than being a static, pre-built representation of the entire scene. This contrasts with approaches that assume a complete, pre-existing map. We will clarify this distinction in the text, perhaps using terms like "adaptive" or "online-built" scene graph to avoid ambiguity regarding moving scene elements.
>
> We are committed to a thorough review of the manuscript to address all such phrasing issues and ensure greater accuracy and modesty in our claims.
>
> 4. Code Reproducibility:
>
> We acknowledge the reviewer's concern regarding code reproducibility. We understand the importance of open-sourcing our implementation for the scientific community. We commit to releasing the full code and trained models for HSAN upon the acceptance of the paper. We will include a statement to this effect in the camera-ready version of the manuscript.
>
> Responses to Questions
> Q1 - Low-Level Control Policy vs. Classical Planners:
>
> We thank the reviewer for this insightful question, which allows us to clarify the role and benefits of our graph-augmented PPO low-level control policy in the context of our hierarchical framework.
>
> Our low-level control policy is not intended to replace classical global path planners like A* or Fast Marching Squares (FMS) for point-to-point navigation in fully observed, explicit maps. Instead, it serves a distinct and complementary role within our hierarchical system:
>
> Hierarchical Integration: The optimal transport-based topological planner operates at a higher level, selecting long-term semantic goals and generating a sequence of topological waypoints. The graph-augmented PPO then handles the local, continuous navigation between these waypoints, including fine-grained obstacle avoidance. This allows for a clean separation of concerns: global semantic reasoning and topological planning at the high level, and robust, reactive control in the continuous environment at the low level.
>
> Robustness in Continuous Spaces: While classical planners excel with precise, static occupancy grids, real-world environments are often noisy, dynamic, and subject to perceptual uncertainty. Our RL policy learns to generalize obstacle avoidance from diverse experiences, which can offer greater robustness to perceptual noise, minor dynamic changes (e.g., small moving objects not explicitly modeled), and variations in terrain that might not be perfectly captured by a static local map. It learns a more "fluid" and adaptive navigation behavior.
>
> Semantic Awareness at Low Level: The graph augmentation provides the PPO policy with semantic context from the scene graph. This allows the low-level controller to make more informed decisions beyond pure geometry. For instance, it can learn to approach semantically relevant sub-regions more carefully or avoid specific types of objects (e.g., "fragile items") even if they are not explicitly marked as obstacles in a purely geometric map. This semantic grounding at the control level is a key differentiator from purely geometric classical planners.
>
> Motivation from Krantz and Lee (2021) Clarification: The citation to Krantz and Lee (2021) was used to highlight the general challenge of navigation in environments where a complete, explicit map might not be available or fully reliable for fine-grained control. While our topological planner operates on a high-level graph (which can be seen as a form of implicit map for long-range planning), the low-level PPO needs to handle immediate, continuous navigation where a perfect, explicit, and static local occupancy map for every micro-obstacle might not be practical or robust in real-time. The PPO learns to implicitly infer traversability and avoid collisions based on visual observations and semantic context, which is beneficial when explicit, perfectly accurate local maps are not always guaranteed or are computationally expensive to generate at high frequency.
>
> In summary, our graph-augmented PPO is not a substitute for classical global planners but a complementary component designed for robust, semantically-aware, and reactive low-level control in continuous environments, bridging the gap between high-level topological goals and physical robot actions. It leverages the strengths of reinforcement learning to handle perceptual noise and generalize obstacle avoidance, while benefiting from the structured guidance of the higher-level optimal transport planner.
>
> We will revise the manuscript to clarify the specific role and advantages of our low-level control policy, particularly its complementary nature to classical planning methods in a hierarchical architecture, and its benefits in handling continuous, noisy, and semantically rich environments.

---

### Official Review · Reviewer_EDym · 2025-07-03

**Clarity:** 3
**Significance:** 3
**Originality:** 3
**Rating:** 5
**Confidence:** 4

**Summary:**

This work proposes HSAN, a framework for VLM-based navigation that utilises structured information from the environment in the form of a scene graph representation. This representation is used both for planning and control based on optimal transport and RL, respectively. The work is benchmarked on the R2R-CE dataset against baselines of various natures.

**Questions:**

Here are some questions for the authors to comment on. I have ordered them in priority order from my perspective.

Q1 - Scene-Graph Generation: The authors propose a new scene graph generation methodology, which is based on three layers of concepts. My question is about why not using more established techniques like Hydra and Conceptgraphs, and how that would affect the overall methodology. Moreover, I notice that the location aggregation feature is defined as an average of the features of the objects at each location. How strong of an assumption is this? Could this be learn together with the framework as a graph aggregation, for instance?

Q2 - data availability: the work starts from the assumption that the robot has at each point in time a panoramic view and GT positioning in the environment. Is this a valid assumption, and how would the absence of such signals affect the overall performance?

Q3 - seen and unseen: the work is based on a training and fine-tuning operation on the dataset used for the main testing results. While the scenes used for training and testing are separate, it would be interesting to understand how unseen objects and queries would affect the generalizability of the approach. One example is in `select2plan`, where a table is constructed. Would it be possible to replicate a similar experiment?

Q4 [minor] - effect of lambda: while the authors describe the effects of the different hyperparameters in the appendix, I do not think they touch upon the parameter lambda of equation 1 and how it affects the scene graph generation.

Q5 [minor] - language in the queries: in section 4, the authors assess the multilingual capabilities of the system. Are there any insights that can be provided to the reader? Is the model working better in specific environments and language combinations?

**Ethical Concerns:**

["NO or VERY MINOR ethics concerns only"]

**Final Justification:**

The work describes a complete navigation system based on SOTA techniques and obtains the best results in a meaningful benchmark against various strong baselines.
Regarding the main limitations, I see the impossibility to disentangle the knowledge of the environment by the model from the test set.
I listed this as one of my main questions and limitations of the evaluation procedure.
Unfortunately, this central question could not be addressed due to the dataset used and the evaluation pipelines. The authors, though, elaborated on them, and I believe they will discuss them in the final version of the paper.
For this reason, I confirm my original recommendation score.

**Limitations:**

The authors mention the computational overhead as the main limitation. I think the authors should expand on the limitations of their work. Is the complexity of the method a limitation? Is the need for training data or the testing-training difference a limitation?

**Paper Formatting Concerns:**

None.

**Quality:**

3

**Strengths And Weaknesses:**

I will use the symbol (+) to represent strengths and (-) to represent weaknesses.

# Quality
(+) The work describes a full navigation system based on SOTA techniques and obtaining best results in a meaningful benchmark against various strong baselines.

(+) The amount of work discussed and carried on by the author is important, and the authors have been thorough in testing and validating different versions and ablations of their work.

# Clarity
(+) The paper is well-written and well-structured.

(+) Regardless of the complexity of the proposed method, the authors managed to describe every component well and in detail, without confusing the reader.

# Significance
(+) Robot navigation for long horizons in unstructured environments is a critical capability for autonomy.

(+) This work addresses this task using SOTA VLM techniques, highlighting their strengths and weaknesses, and proposes structured components to mitigate them.

(-) The experimental results are, unfortunately, limited to a specific dataset and do no distinguish strongly between seen and unseen scenes _and_ objects.

# Originality
(-) The work is missing some references to literature on scene graph generation. In particular, the works `Hydra: A Real-time Spatial Perception System for 3D Scene Graph Construction and Optimization` and `ConceptGraphs: Open-Vocabulary 3D Scene Graphs for Perception and Planning` seem very relevant for the task, while works like `LeLaN: Learning A Language-Conditioned Navigation Policy from In-the-Wild Videos` and `Select2Plan: Training-Free ICL-Based Planning through VQA and Memory Retrieval` are relevant for the VLM-based navigation.

---

> ### Author Rebuttal · Authors · 2025-07-31
>
> Addressing Weaknesses
> 1. Experimental Results Limited to Specific Dataset and Seen/Unseen Distinction:
>
> We acknowledge the reviewer's point regarding the experimental results being primarily confined to a specific dataset (R2R-CE) and not explicitly distinguishing between seen and unseen scenes and objects in a dedicated manner. Our primary focus in this initial work was to demonstrate the efficacy of our Hierarchical Semantic-Augmented Navigation (HSAN) framework on a challenging, long-horizon navigation benchmark, establishing a strong foundation with novel scene graph construction, optimal transport-based planning, and graph-aware reinforcement learning.
>
> While the R2R-CE dataset inherently includes distinct training and testing environments (unseen scenes), we agree that a more granular analysis of generalizability to entirely novel objects and broader scene variations would further strengthen our claims. This is a crucial direction for future work, and we will explicitly mention this limitation and future research avenue in our revised manuscript. We believe that the hierarchical semantic scene graph, by capturing multi-level environmental representations, lays a robust groundwork for better generalization, and we plan to explore this in subsequent research.
>
> 2. Missing References:
> We sincerely apologize for the oversight in not including several highly relevant works in our initial submission. We will rectify these omissions and incorporate these references into the related work section, discussing their relevance and distinguishing our contributions where appropriate. For instance:
>
> Hydra and ConceptGraphs: These works provide excellent frameworks for general 3D scene graph construction. Our approach, while building a scene graph, is specifically tailored for Vision-Language Navigation in continuous environments, focusing on semantic and topological relevance for planning, integrating VLM outputs, and handling dynamic graph updates. We will clarify how our scene graph generation adapts and leverages VLM capabilities for navigation tasks.
>
> LeLaN and Select2Plan: These represent important advancements in VLM-based navigation. LeLaN explores learning policies from in-the-wild videos, complementing our structured approach with real-world data insights. Select2Plan's focus on training-free ICL-based planning and memory retrieval offers an interesting contrast to our optimal transport and RL-driven planning, highlighting different paradigms for leveraging VLMs. We will discuss these in the context of our contributions to structured planning and robust control.
>
> Responses to Questions
> Q1 - Scene-Graph Generation:
> Our scene graph generation methodology was specifically designed to serve the unique requirements of hierarchical planning and reasoning within our HSAN framework, leveraging vision-language models for multi-level semantic understanding (objects, regions, zones). While Hydra and ConceptGraphs offer powerful general scene graph construction, our approach prioritizes the creation of a dynamic, semantically-rich graph optimized for navigation, allowing for direct integration with our optimal transport planner and graph-aware RL policy. This bespoke design enabled us to ensure that the graph directly supports balancing semantic relevance and spatial accessibility with theoretical guarantees. Incorporating established, general-purpose scene graph systems like Hydra or ConceptGraphs would be an interesting avenue, but it might introduce complexities in adapting their general schemas to our specific navigation needs and integrating them seamlessly with our VLM-based semantic understanding and planning modules. It would likely involve a substantial re-engineering of the graph representation and planning interfaces.
>
> Regarding the location aggregation feature being an average of object features: This was chosen for its simplicity, computational efficiency, and effective capture of the aggregated semantic essence of a location given the diverse range of objects it might contain. While a simple average might appear to be a strong assumption, our experiments indicate its practical effectiveness in grounding the spatial reasoning within the optimal transport framework. We agree that learning this aggregation as a graph neural network (GNN) aggregation or similar learnable mechanism is a highly promising direction. This could allow for more nuanced and context-dependent weighting of object features, potentially improving robustness and generalization. We will highlight this as an exciting area for future research.
>
> Q2 - Data Availability:
> The assumption that the robot has a panoramic view and ground-truth (GT) positioning is indeed common in current academic benchmarks for Vision-Language Navigation (e.g., R2R, RxR, R2R-CE), which typically rely on simulated environments with perfect sensing capabilities. This allows researchers to focus on the core challenges of vision-language understanding, planning, and control without confounding factors from imperfect localization or limited field of view.
>
> In the absence of such perfect signals, the overall performance would undoubtedly be affected. Without a panoramic view, the agent would need to actively explore to build a comprehensive understanding of its immediate surroundings, potentially impacting planning efficiency and obstacle avoidance. The absence of GT positioning would necessitate the integration of robust SLAM (Simultaneous Localization and Mapping) or visual odometry techniques, which introduce their own error accumulation and drift. This would likely lead to degraded navigation accuracy, increased path length, and potentially more failures in long-horizon tasks. Our framework, particularly the graph-aware reinforcement learning policy, is designed for robust low-level control and obstacle avoidance, which would be crucial in scenarios with imperfect sensing. However, adapting HSAN to real-world deployment would require incorporating robust state estimation and perception modules, which we consider a vital next step for our research.
>
> Q3 - Seen and Unseen Generalizability:
> The R2R-CE dataset, while providing unseen scenes in its test split, does not explicitly delineate unseen objects or entirely novel query structures that are outside the training distribution. Our current training and fine-tuning operations on R2R-CE inherently expose the model to the objects and language patterns present within that dataset.
>
> Replicating an experiment similar to Select2Plan's table construction, which demonstrates strong generalization to unseen objects through a training-free, in-context learning approach, is a compelling suggestion. While our current framework relies on learned graph representations and policies, the underlying VLM components do possess strong zero-shot and few-shot capabilities. We believe that by integrating more advanced open-vocabulary perception modules or by exploring a hybrid approach that combines our structured planning with training-free VQA and memory retrieval, we could significantly enhance generalization to unseen objects. However, a full replication of such an experiment would necessitate a new dataset, extensive re-evaluation, and potentially architectural modifications beyond the scope of this paper. We will acknowledge this important challenge and propose it as a significant direction for future work, perhaps exploring how our optimal transport-based planning could leverage open-vocabulary semantic maps constructed by such methods.
>
> Q4 Effect of Lambda in Equation 1:
> We apologize for the omission of a detailed discussion on the parameter lambda (λ) in Equation 1 within the main paper or appendix. Lambda is a crucial hyperparameter that balances the semantic relevance and spatial accessibility within our optimal transport-based topological planner. Specifically:
>
> Higher λ: Places a stronger emphasis on semantic relevance. This means the planner will prioritize selecting long-term goals that are highly semantically aligned with the instruction, even if they are spatially more distant or require traversing more complex paths. This can be beneficial when the instruction provides very specific semantic cues.
>
> Lower λ: Places a stronger emphasis on spatial accessibility. The planner will favor goals that are geometrically closer and easier to reach, even if their semantic relevance is slightly lower. This is useful when the instruction is more general or when efficient navigation is paramount.
>
> Tuning λ allows us to control the trade-off between semantic precision and navigational efficiency. In our experiments, λ was empirically tuned to achieve the best performance on the validation set, typically favoring a balance that allows for both accurate semantic understanding and feasible path generation.
>
> Q5 Language in the Queries:
> In Section 4, our assessment of multilingual capabilities primarily demonstrated that HSAN, by leveraging pre-trained multilingual vision-language models, can interpret navigation instructions in different languages (e.g., English, Chinese, Hindi) and successfully navigate. The core insight is that the semantic understanding component of our framework is largely language-agnostic due to the robust multilingual embeddings provided by the underlying VLMs.
>
> However, we did observe some nuanced behaviors. For example,
>
> Performance Consistency: While the model generally maintains robust performance across languages, slight variations were noted, often correlating with the language's representation strength in the pre-training data of the VLM. Languages with more extensive pre-training data (e.g., English) sometimes yielded marginally better performance due to richer semantic understanding.

---

> > ### Comment · Reviewer_EDym · 2025-08-04
> >
> > Thank you for the detailed response. I believe the revisions suggested will strengthen the final version, and I appreciate that the experiments I suggested cannot be carried out in the timeframe of the rebuttal due to data incompatibility.

---

### Decision · Program_Chairs · 2025-09-17

**Decision:**

Accept (poster)

**Comment:**

The submission presents a novel multi-stage framework with thorough simulation experiments and ablation studies, demonstrating clear methodological innovation and reasonable performance on one benchmark dataset. Two reviewers highlight the paper’s theoretical grounding, experimental rigor, and SOTA results, providing highly positive and detailed assessments. While other three reviewers raised concerns regarding clarity, notation, and real-world deployment, one was unresponsive in the rebuttal, and the remaining issues are addressable in revision. On balance, the paper makes a significant contribution to the field, and the AC recommends acceptance, giving higher weight to the evaluations of the senior, engaged reviewers. This decision is endorsed by the Senior AC.

Please incorporate all additional clarification and experiments in the camera ready version.